# Hair Growth Promotion by Extracts of Inula Helenium and Caesalpinia Sappan Bark in Patients with Androgenetic Alopecia: A Pre-clinical Study Using Phototrichogram Analysis

**Hyoung Chul Choi [1,2], Gae Won Nam [3], Noh Hee Jeong [2,\*] and Bu Young Choi [4,\*]**

1   Cosmecutical R&D Center, HP&C, Cheongju 28158, Korea; hcchoi@hpnc.co.kr
2   Department of Engineering Chemistry, Chungbuk National University, Cheongju 28644, Korea
3   Department of Cosmetics, Seowon University, Cheongju 28644, Korea; skarod@seowon.ac.kr
4   Department of Pharmaceutical Science & Engineering, Seowon University, Cheongju 28644, Korea
\*   Correspondence: nhjeong@chungbuk.ac.kr (N.H.J.); bychoi@seowon.ac.kr (B.Y.C.);
    Tel.: +82-043-261-2440 (N.H.J.); +82-043-299-8411 (B.Y.C.)

**Abstract:** *Inula helenium* (IH) is known to possess antifungal, anti-bacterial, anti-helminthic, and anti-proliferation activities. *Caesalpinia Sappan* (CS) is known to reduce inflammation and improve blood circulation. Based on their folkloric use, these plants are expected to be promising candidates for promoting hair growth and preventing hair loss. Moreover, these plants are rich sources of certain phytochemicals, which have been reported to promote hair growth. In this clinical trial, we investigate the efficacy of a scalp shampoo formulated by mixing extracts of IH and CS in preventing hair loss and promoting hair growth in patients with androgenetic alopecia. Using a phototrichogram (Folliscope 2.8, LeadM, Korea), we compared the hair density and total hair counts in patients receiving the scalp shampoo at baseline, and at 8, 16, and 24 weeks after use of the shampoo. We found a statistically significant increase in the total hair count in the test group (n = 23) after 16 and 24 weeks of using the scalp shampoo (2.17 n/cm$^2$ ± 5.72, $p < 0.05$; and 4.30 n/cm$^2$ ± 6.37, $p < 0.01$, respectively) as compared to the control subjects. Based on the results of this clinical study, we conclude that the IH and CS extract complex is a promising remedy for preventing hair loss and promoting hair growth.

**Keywords:** *Inula helenium*; *Caesalpinia Sappan*; hair growth; phototrichogram; alopecia

## 1. Introduction

Hair loss is a common disorder that affects people of all ages [1]. The most common type of hair loss is androgenetic alopecia, which is characterized by the formation of very fine vellus hairs resulting from the miniaturization of the hair follicles, thereby leading to the appearance of baldness [2]. Currently, available treatment strategies for alopecia include autologous transplantation of hair follicles and therapeutic intervention using drugs that promote hair growth. Only two Food and Drug Administration (FDA)-approved hair growth-promoting drugs, namely minoxidil and finasteride, are available in the market [3]. However, these drugs are of limited use because of adverse drug reactions, such as skin irritation, sexual dysfunction, and circulatory disorders [4]. Moreover, certain steroidal drugs commonly used to treat persistent dandruff and scalp inflammation, the conditions that contribute to patterned hair less, may have adverse effects if used for prolonged periods [5]. Considering the psychosocial impacts of hair loss and the adverse effects of current therapies, there have been growing interest in developing new therapies to treat alopecia. Over the centuries, a wide range of plant extracts and formulated herbal products have been used traditionally as hair tonics in many societies. Thus,

systematic scientific research to evaluate the hair growth-promoting potential of alternative remedies, such as herbal products, is a rational approach for developing new therapies for hair loss.

The hair growth cycle comprises of anagen, catagen, and telogen phases [6]. During the transition between these phases, a group of specialized fibroblasts known as dermal papillae cells (DPCs) present in the hair follicle bulb plays an essential role in the regulation of the hair growth cycle [7]. Therefore, factors affecting the functions of DPCs are of great importance for the development of therapy for the treatment and/or prevention of hair loss [8]. These factors include, but not limited to, multiple signaling molecules, such as Wnts, Sonic hedgehog (Shh), and transforming growth factor-beta (TGF-β), which contribute to the anagen initiation of multipotent epithelial stem cells [9]. Targeting these multi-component biochemical signaling pathways of hair growth regulation would be a rational approach for developing novel therapeutics for the treatment of alopecia [10].

We have recently reported that costunolide, a sesquiterpene lactone present in the roots of *Inula helenium* L. (Asteraceae), promotes hair growth in vivo, likely by activating intracellular signaling pathways including the Wnt/β-catenin, Shh/Gli, and TGF-β/Smad signaling pathways, which are associated with cell cycle regulation in hair follicular cells [10]. Moreover, 3-deoxysappanchalcone (3-DSC), a sesquiterpene lactone present in *Caesalpinia Sappan* L. (Leguminosae), has been shown to promote proliferation of dermal papillae and stimulate hair growth partly via activation of Wnt/β-catenin signaling and inhibition of signal transducer and activator of transcription-6 (STAT6)-mediated quiescence of hair follicular cells [11]. However, a clinical trial investigating the effect of extracts of *Inula helenium* (IH) and *Caesalpinia Sappan* (CS) on hair growth has not been performed yet. The purpose of this study is to evaluate the effect of IH and CS extract complex on the prevention and treatment of hair loss and the enhancement of scalp hair growth in healthy subjects.

## 2. Materials and Methods

### 2.1. Materials

A newly formulated scalp shampoo (HPNC-RX-G-001) was prepared by adding 0.3% of IH extract and 0.1% of CS bark extract to the shampoo. The control product (HPNC-RX-B-001) was a scalp shampoo that lacked IH extract and CS bark extract but contained all other ingredients in the same quantity as that of HPNC-RX-G-001. The materials were provided to the patients in identical containers with a corresponding randomization number.

### 2.2. Subjects and Study Design

Healthy Korean subjects (n = 20), both males and females, ranging from 18 to 54 years of age, were selected for the study on the basis of inclusion and exclusion criteria. Following the basic and specific (BASP) classification [12], male and female subjects (18 to 54 years of age) with androgenic alopecia were selected. Subjects diagnosed with more than M2, C2, and U1 range in the basic type and more than the V1, F1 range in the specific type according to BASP classification were included. Exclusion criteria included participation in a previous study within the last 6 months, prior surgical correction of scalp hair loss, use of topical minoxidil or finasteride within 6 months; other skin diseases of the scalp, including severe seborrheic dermatitis, psoriasis, lichenoid eruption, or other scalp infections. Alterations in hair styling and dyeing of the hair were restricted during the study.

The study of scalp shampoo was approved by the Institutional Review Board (IRB) of the local committee and was conducted on 29 June 29 2018. All subjects gave written informed consent prior to participating in the study and applied about 3 mL of the assigned product twice a day in the morning and evening. The scalp shampoo was massaged for 5 minutes with warm water followed by rinsing with water. Product usage and compliance were monitored for each participant by reviewing product-use-diaries maintained by each participant and weighing the returned test products at each scheduled visit (Weeks 8, 16, and 24).

*2.3. Measurement of Hair Density and Total Hair Counts*

All participants placed their head onto a hair photograph device(SnT Lab, Seoul, Korea), and photographs were taken at a fixed distance, angle, and lighting with a digital camera (Nikon D 300, Tokyo, Japan). A global evaluation was performed by comparing clinical images obtained at the baseline, and at 8-, 16-, and 24-weeks after treatment with the test products. Blinded investigators assessed clinical images on a 7-point scale (−3, marked decrease; −2, intermediate decrease; −1, slight decrease; 0, no change; +1, slight increase; +2, intermediate increase; and +3, marked increase).

An evaluation area of 1 cm$^2$ was clipped with approximately 2 mm length after a red spot was tattooed at the site of hair loss (vertex or forehead hairline). Total hair count was measured using a phototrichogram system, Folliscope(ver 2.8, Lead M, Seoul, Korea), and analyzed at the baseline and 16 and 24 weeks after treatment with the test products. Total hair count (number/cm$^2$) was calculated within an area of 1 cm$^2$ [13].

## 3. Statistical Analysis

Data are expressed as mean ± SD of the differences in the changing rate between the baseline value and the value obtained at each time point of using the scalp shampoo. All statistical analyses were performed using the SPSS® package program (SPSS Inc, Chicago, IL, U.S.A.). For analysis of the hair density, statistical analysis using the repeated measures ANOVA ($p < 0.05$) was performed taking variables from different time points or groups. For analysis of total hair count, the significance of a difference between time points was calculated using the paired *t*-test considering $p < 0.05$ as the significant level. A questionnaire survey was conducted on the efficacy of the testing product after 8, 16, and 24weeks treatment and the usability was conducted after 24 weeks of treatment. The assessment rated each criterion on a scale of one to seven (1: strongly disagree, 2: disagree, 3: somewhat disagree, 4: no change, 5: somewhat agree, 6: agree, 7: strongly agree).

## 4. Results

The stages of the study with the number of subjects are summarized in the study flowchart (Figure 1). A total of 50 male and female patients with moderate androgenetic alopecia were recruited and randomly assigned to either the test group (n = 25) or the control group (n = 25). Three subjects dropped out of the study because of a withdrawal of consent (n = 3).

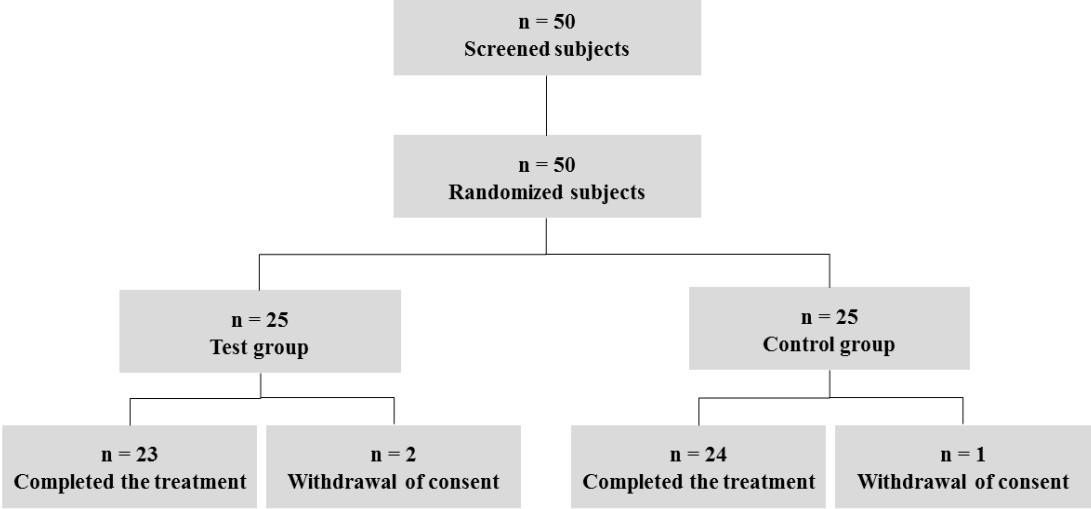

**Figure 1.** Flowchart of the clinical trial and subject disposition. A total of 50 male and female patients with moderate androgenetic alopecia were recruited and randomly assigned to either the test group (n = 25) or the control group (n = 25). Three subjects (n = 3) dropped out of the study because of a withdrawal of consent.

### 4.1. General Characteristics of the Subjects

Of the total 25 subjects under the test group, two subjects were dropped from the study. Finally, 23 subjects completed the study. The mean age of the participants was 40.33 ± 7.88 years, and the ratio of males to females was 7:16 in the test group. In the control group, 24 subjects participated in the study. The mean age of the participants was 41.08 ± 6.74 years, and the ratio of males to females was 6:18. The demographic information of each group is shown in Table 1, and no statistically significant differences in demographic settings were found between the two groups.

**Table 1.** Demographic characteristics of the subjects.

| Characteristics | | Test Group n = 23 | Control Group n = 24 | *p*-Value |
|---|---|---|---|---|
| | | n (%) | n (%) | |
| **Sex** | Male | 7 (30.43) | 6 (25.00) | |
| | Female | 16 (69.57) | 18 (75.00) | |
| **Age** | **Mean ± SD** | **40.33 ± 7.88** | **41.08 ± 6.74** | 0.448 |
| | Median | 42.00 | 43.00 | |
| | Min, Max | 24.00, 52.00 | 25.00, 51.00 | |
| | 20–29 | 3 (13.04) | 2 ( 8.33) | |
| | 30–39 | 6 (26.09) | 6 (25.00) | |
| | 40–49 | 12 (52.17) | 15 (62.50) | |
| | ≥50 | 2 ( 8.70) | 1 ( 4.17) | |

### 4.2. Effects of Scalp Shampoo on Hair Density and Total Hair Counts

Compared to baseline, the photo assessment of hair density showed an increasing trend in test groups (Figure 2, Table 2a). Phototrichogram analysis revealed that the total hair count was significantly increased in the test group at 16 and 24 weeks after treatment as compared to the baseline ($p < 0.05$, $p < 0.01$) (Table 2b). The degree of change in total hair count between the baseline and after 16 weeks of treatment was 2.17 ± 5.72 number/cm$^2$, and that between baseline and 24 weeks of treatment was 4.30 ± 6.37 number/cm$^2$.

**Table 2.** Descriptive statistical analysis of hair density (**a**) and total hair counts (**b**) between time points.

| (a) | | Baseline | 8 weeks | 16 weeks | 24 weeks |
|---|---|---|---|---|---|
| Control group (n = 24) | Mean [1] | 0.00 | 0.00 | 0.04 | 0.00 |
| | SD | 0.00 | 0.00 | 0.20 | 0.29 |
| | *p*-value | - | 1.000 | 0.317 | 1.000 |
| Test group (n = 23) | Mean [1] | 0.00 | 0.04 | 0.09 | 0.13 |
| | SD | 0.00 | 0.21 | 0.29 | 0.34 |
| | *p*-value | - | 0.317 | 0.157 | 0.083 |
| (b) | | Baseline | 8 weeks | 16 weeks | 24 weeks |
| Control group (n = 24) | Mean (n/cm$^2$) [2] | 133.75 | - | 135.29 | 134.71 |
| | SD | 20.54 | - | 21.45 | 21.34 |
| | *p*-value [3] | - | - | 0.085 | 0.285 |
| Test group (n = 23) | Mean (n/cm$^2$) [2] | 145.70 | - | 147.87 | 150.00 |
| | SD | 23.82 | - | 23.21 | 22.83 |
| | *p*-value [3] | - | - | 0.044 * | 0.003 ** |

[1] Increment of mean value represents an improvement of hair condition on vertex and hairline. [2] Increment of mean value represents an improvement of hair counts; n stands for "number". [3] Significantly different at * $p < 0.05$, ** $p < 0.01$ compared with baseline.

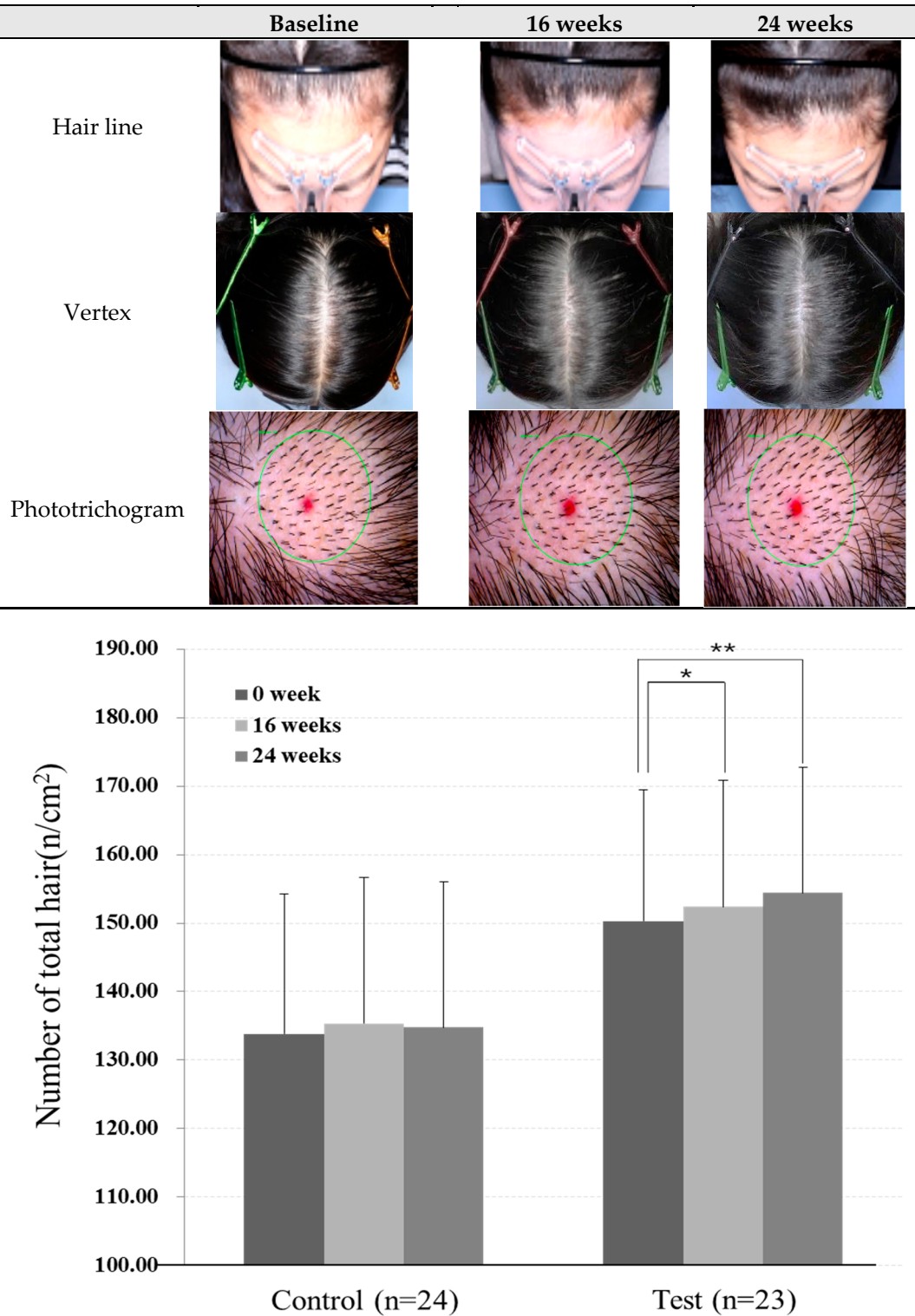

**Figure 2.** Images of change in hair pattern following the application of scalp shampoo for 24 consecutive weeks. Compared to the baseline data, photographic assessment of hair density showed a tendency of increase in hair density in test groups. Bottom panel bar graph represents the change of total hair counts measured by folliscope image at baseline and after the application of scalp shampoo (means ± SD, * $p < 0.05$ 16 weeks vs. baseline, ** $p < 0.01$ 24 weeks vs. baseline).

### 4.3. Assessment of Self-Questionnaires by Subjects

As a result of self-questionnaires for efficacy, the positive answer rate of the test group about "improvement of frontal hairline" was significantly higher than that of the control group at 24 weeks after treatment ($p < 0.05$, Figure 3 and Table 3).

**Table 3.** Statistical analysis of self-questionnaires for efficacy between test and control groups.

| Group | Item | *p*-Value [1] | | |
|---|---|---|---|---|
| | | **8 weeks** | **16 weeks** | **24 weeks** |
| **Control vs. Test** | Improvement of vertex hair | 0.706 | 0.207 | 0.230 |
| | Improvement of frontal hairline [1] | 0.524 | 0.574 | **0.025 *** |
| | Improvement of hair loss | 0.703 | 0.537 | 0.119 |
| | Hair satisfaction | 0.715 | 0.238 | 0.140 |

[1] Significantly different at * $p < 0.05$ compared with the control group.

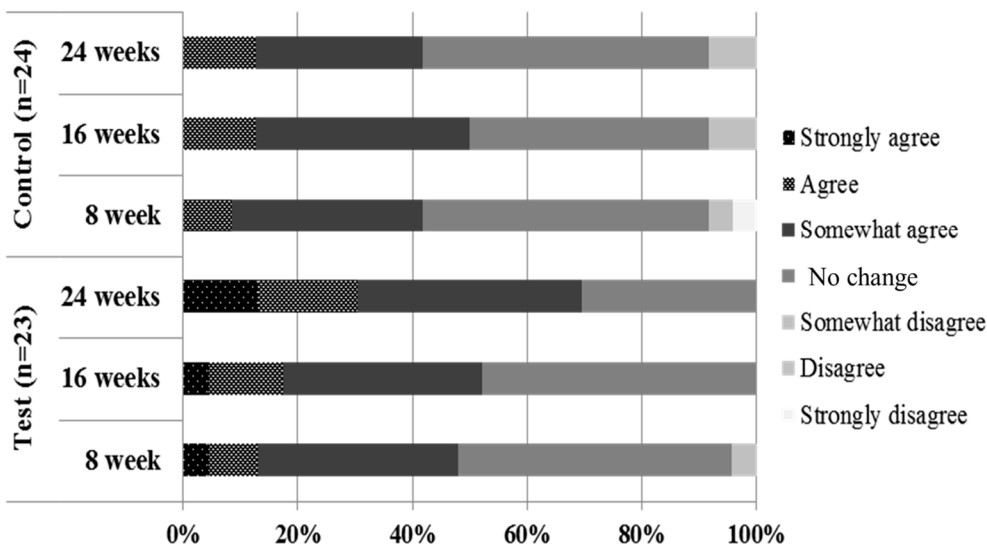

**Figure 3.** Results of answers in self-questionnaires for efficacy (improvement of frontal hairline). As a result of self-questionnaires for efficacy, the positive answer rate of the test group about "improvement of frontal hairline" was significantly higher than that of the control group at 24 weeks after treatment.

## 5. Discussion

Hair loss is a common disorder that affects men, women, and children due to aging, diseases, and exposure to various environmental toxicants and medications [1]. The most common type of hair loss is androgenetic alopecia, where the miniaturization of the hair follicles results in the formation of very fine vellus hairs, giving the appearance of a bald scalp [2]. Alopecia resulting from progressing hair loss needs attention to develop therapies for correcting deregulated pathophysiological conditions, thereby improving the psychosocial status of the affected individuals. This clinical trial was designed to evaluate the efficacy of a formulated scalp shampoo among volunteers with androgenic alopecia. The formulated scalp shampoo was made by mixing IH extract and CS extract, both of which have been reported to possess hair growth-promoting constituents, such as costunolide and 3-deoxysappanchalcone, respectively [10,11,14]. As compared to the baseline hair count, the statistically significant increase in the hair density and total hair counts after 16 and 24 weeks of using scalp shampoo among the study subjects suggests that the IH and CS extract complex assisted in preventing hair loss and promoting hair growth.

Since the increase in hair density and hair count was more profound at 24 weeks of application of the scalp shampoo as compared to 16 weeks of use, it may be conferred that the continuous application of the product maximizes its effects. The overall improvement in hair growth among the test subjects compared to the control group suggests that the hair shampoo formulations containing IH and CS extracts are effective for treating patients with androgenetic alopecia.

Although the mechanistic basis of hair growth promotion by the formulated scalp shampoo has not yet been examined, a previous study from our laboratory has demonstrated the presence of hair growth-promoting agents, such as costunolide and 3-deoxysappanchalcone (3-DSC) as major bioactive principles from IH and CS extracts, respectively. We have reported that these major bioactive compounds, when applied separately, modulated various biochemical pathways, thereby leading to hair growth in vivo. Whereas costunolide promotes hair growth in vivo, likely by activating intracellular signaling pathways associated with cell cycle regulation in hair follicular cells including the Wnt/β-catenin, Shh/Gli, and TGF-β/Smad signaling pathways [10], 3-DSC promotes proliferation of dermal papillae and stimulates hair growth partly via activation of Wnt/β-catenin signaling and inhibition of STAT6-mediated quiescence of hair follicular cells [11]. Thus, the new scalp shampoo formulation containing IH and CS extracts is enriched with both hair growth-promoting phytochemicals that have a diverse mode of action and the product effectively prevents hair loss and promotes hair growth. No other remarkable adverse reactions were observed during the course of this study. To the best of our knowledge, this study is the first clinical trial to analyze the feasibility of shampoo containing IH and CS extract complex. This study lays important groundwork for designing further studies with a larger sample size to validate the efficacy and ensure the safety of the product, and to elucidate the underlying mechanisms of action.

**Author Contributions:** Conceptualization, H.C.C. and N.H.J.; methodology, G.W.N.; software, G.W.N.; validation, N.H.J. and B.Y.C.; formal analysis, H.C.C. and G.W.N.; investigation, H.C.C.; resources, H.C.C. and G.W.N.; data curation, G.W.N.; writing—original draft preparation, H.C.C.; writing—review and editing, N.H.J. and B.Y.C.; visualization, N.H.J. and B.Y.C.; supervision, N.H.J. and B.Y.C.; project administration, H.C.C.

**Funding:** This research received no external funding.

**Acknowledgments:** We wish to thank Joydeb Kumar Kundu for helpful comments on the manuscript.

**Availability of Data:** The data that support the findings of this study are available from the corresponding author on request.

**Conflicts of Interest:** The authors declare no potential conflict of interest.

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
