# Peer review of "Hair Growth Promotion by Extracts of Inula Helenium and Caesalpinia Sappan Bark in Patients with Androgenetic Alopecia: A Pre-clinical Study Using Phototrichogram Analysis"

_cosmetics, doi:10.3390/cosmetics6040066_

Round 1
Reviewer 1 Report
it is a well written paper and nice study but I don't know how a shampoo permits improving the hair loss with a so great result
Author Response
Author's Reply to the Review Report (Reviewer 1)
Thanks a lot for your delicate comment
Reviewer #1: it is a well written paper and nice study but I don't know how a shampoo permits improving the hair loss with a so great result.
Author reply: Thank you for your interest and nice comment. A newly formulated scalp shampoo is designed to reduce hair loss symptoms and contains herbal extracts that help to improve hair growth. This shampoo enriched with plant extract that nourishes and soothes damaged scalp and hair to prevent hair loss and strengthen the weakened hair roots. The excellent cleansing capacity of shampoo thoroughly removes dead skin cells from the scalp for improving the scalp and hair condition to keep it healthy. The Shampoo is the form of the product to apply but it contains chemicals from plant extracts that promote hair growth by activating dermal papillary cells proliferation via modulation of cellular signaling pathways. Further studies are warranted to delineate the molecular mechanisms underlying the efficacy of the product in hair loss prevention.

Reviewer 2 Report
The number of the pictures is not reportes (lines 136 - 137)
Keywords: please adjust "Phototrichogram"
Materials and Methods:
If I have understood weel, the researchers used a shampoo to evaluate the efficacy of the two ingredients. Indeed, the use of a shampoo twice a day is not a real condition. The application of 5 minutes massage again is not a real condition. It seems to be more similar to the use of a lotion. Could you clarify?
The bibliography number 12 is not related to the BASP classification. Could you clarify?
Line 101: the measurment with the Folliscope has been taken at baseline and after 24 weeks. Indeed, the pictures of Phototrichogram (before Figure 2) show also a check at 16 weeks. Could you clarify?
Line 105-111: the researcher reports the statistical evaluation of the hair density and hair counts. Nothing is told about the statistical evaluation of the self questionnaire
Author Response
Author's Reply to the Review Report (Reviewer 2)
Thank you very much for thorough and careful reviewing of the manuscript. We appreciate your valuable comments
Reviewer #2: The number of the pictures is not reportes (lines 136 - 137)
Keywords: please adjust "Phototrichogram"
Author reply:
adjusted : “Photopictogram” to “Phototrichogram”
Reviewer #2: Materials and Methods:
If I have understood well, the researchers used a shampoo to evaluate the efficacy of the two ingredients. Indeed, the use of a shampoo twice a day is not a real condition. The application of 5 minutes massage again is not a real condition. It seems to be more similar to the use of a lotion. Could you clarify?
Author reply:
The basic way to use shampoo is to use regular scalp once a day, but it is recommended that people with high levels of exposure to scalp due to oily hair or occupational conditions use it twice in the morning and evening. Since a newly formulated scalp shampoo is designed to reduce hair loss symptoms, it is recommended to use shampoo twice a day with massage for 5 minutes and wash to get sufficient effect. Also shampoo is made to form a soft and rich foam. The use of Shampoo for hair cleaning and shining of normal people is limited to once a day application, but this product is designed particularly for people having hair loss issue.
The bibliography number 12 is not related to the BASP classification. Could you clarify?
Author reply:
Lee, W.S; Ro, B.I; Hong, S.P; et al. A new classification of pattern hair loss that is universal for men and women: basic and specific (BASP) classification. J Am Acad Dermatol 2007;57:37-46.
Line 101: the measurement with the Folliscope has been taken at baseline and after 24 weeks. Indeed, the pictures of Phototrichogram (before Figure 2) show also a check at 16 weeks. Could you clarify?
Author reply:
adjust : “24 weeks” to “16 and 24 weeks”
Line 105-111: the researcher reports the statistical evaluation of the hair density and hair counts. Nothing is told about the statistical evaluation of the self questionnaire.
Author reply:
Questionnaire survey was conducted on the efficacy of the testing product after 8, 16 and 24weeks treatment and the usability was conducted after 24 weeks treatment. The assessment rated each criterion on a scale of one to seven (1:strongly disagree, 2: disagree, 3: Somewhat disagree, 4: no change, 5: Somewhat agree, 6: agree, 7:strongly agree).
